# AP-3 vesicle uncoating occurs after HOPS-dependent vacuole tethering

Jannis Schoppe[1], Muriel Mari[2], Erdal Yavavli[1], Kathrin Auffarth[1], Margarita Cabrera[3], Stefan Walter[4], Florian Fröhlich[4,5] & Christian Ungermann[1,4,*]

## Abstract

Heterotetrameric adapter (AP) complexes cooperate with the small GTPase Arf1 or lipids in cargo selection, vesicle formation, and budding at endomembranes in eukaryotic cells. While most AP complexes also require clathrin as the outer vesicle shell, formation of AP-3-coated vesicles involved in Golgi-to-vacuole transport in yeast has been postulated to depend on Vps41, a subunit of the vacuolar HOPS tethering complex. HOPS has also been identified as the tether of AP-3 vesicles on vacuoles. To unravel this conundrum of a dual Vps41 function, we anchored Vps41 stably to the mitochondrial outer membrane. By monitoring AP-3 recruitment, we now show that Vps41 can tether AP-3 vesicles to mitochondria, yet AP-3 vesicles can form in the absence of Vps41 or clathrin. By proximity labeling and mass spectrometry, we identify the Arf1 GTPase-activating protein (GAP) Age2 at the AP-3 coat and show that tethering, but not fusion at the vacuole can occur without complete uncoating. We conclude that AP-3 vesicles retain their coat after budding and that their complete uncoating occurs only after tethering at the vacuole.

**Keywords** AP-3; Arf1 GAP; HOPS; vacuole; Vps41
**Subject Category** Membranes & Trafficking
**The EMBO Journal (2020) 39: e105117**

## Introduction

Eukaryotic cells use vesicular transport to shuffle proteins and lipids between organelles of the endomembrane system. Vesicles form at a donor membrane by coat proteins and fuse with their target membrane with the help of a fusion machinery consisting of Rab GTPases, tethering factors, and membrane-embedded SNARE proteins. The generation of vesicles at membranes depends on small GTPases of the Arf1/Sar1 family, which cooperate with distinct coat proteins (Kirchhausen *et al*, 2014). Arf/Sar GTPases require a guanine nucleotide exchange factor (GEF) for their GTP loading and membrane recruitment, and a GTPase activating protein (GAP) to return to the inactive GDP-bound state (Gillingham & Munro, 2007; Donaldson & Jackson, 2011). Transport between ER and Golgi requires the COPII coat with its subunits Sec23-24 and Sec13-31, intra-Golgi, and retrograde transport to the ER use heptameric COPI instead (Gomez-Navarro & Miller, 2016). Transport processes out of the Golgi and toward the endosomal system depend on adapter protein complexes, which according to their discovery have been named AP-1 to AP-5 (Hirst *et al*, 2011). Among these, AP-1 to AP-3 have the most conserved function and have been characterized in molecular detail (Bonifacino, 2014; Kirchhausen *et al*, 2014). All AP complexes have four subunits: two large subunits (e.g. α1 and β1 for AP-1), a medium μ chain, and a small σ subunit. While these complexes have very similar overall structures, they differ in cargo recognition, membrane binding, and clathrin requirements. AP-1 and AP-3 both require Arf1 for membrane binding, whereas AP-2 binds the lipid phosphatidylinositol-4,5-bisphosphate via three distinct sites (Traub *et al*, 1993; Ooi *et al*, 1998; Jackson *et al*, 2010; Ren *et al*, 2013). It has been assumed that AP complexes depend on additional support to form coated vesicles, and both AP-1 and AP-2 indeed require clathrin to form vesicles (Keen & Beck, 1989; Kirchhausen *et al*, 1989).

AP-3 is conserved from yeast to humans (Cowles *et al*, 1997; Dell'Angelica *et al*, 1997; Faundez *et al*, 1998; Odorizzi *et al*, 1998). In yeast, it transports the SNAREs Vam3 and Nyv1, the casein kinase Yck3, and the alkaline phosphatase Pho8 from the Golgi directly to the vacuole (Fig 1A) (Cowles *et al*, 1997; Panek *et al*, 1997; Stepp *et al*, 1997; Darsow *et al*, 1998, 2001; Wen *et al*, 2006). The yeast AP-3 complex consists of the μ3-subunit Apm3 and the σ3-subunit Aps3, as well as the two large subunits, the δ-subunit Apl5, and the β3-subunit Apl6. Unlike the other AP complexes, AP-3 seems to operate efficiently *in vivo* in the absence of clathrin in yeast (Black & Pelham, 2000) and mammalian cells (Zlatic *et al*, 2013), even though clathrin can bind metazoan AP-3 *in vitro* (Drake *et al*, 2000). Some studies thus suggested that AP-3 requires instead the HOPS subunit Vps41 as a coat (Rehling *et al*, 1999; Darsow

1   Department of Biology/Chemistry, Biochemistry Section, University of Osnabrück, Osnabrück, Germany
2   Department of Biomedical Sciences of Cells and Systems, University Medical Center Groningen, University of Groningen, Groningen, The Netherlands
3   Departament de Ciències Experimentals i de la Salut, Universitat Pompeu Farba, Barcelona, Spain
4   Center of Cellular Nanoanalytic Osnabrück (CellNanOs), University of Osnabrück, Osnabrück, Germany
5   Department of Biology/Chemistry, Molecular Membrane Biology Section, University of Osnabrück, Osnabrück, Germany
    *Corresponding author. Tel: +49 541 969 2752; E-mail: cu@uos.de

*et al*, 2001; Asensio *et al*, 2010, 2013; Pols *et al*, 2013). HOPS is a hexameric complex, which is present on vacuoles in yeast and lysosomes in mammalian cells. It functions as a key factor in membrane fusion by tethering late endosomes or autophagosomes to lysosomes/vacuoles and subsequently assembling SNAREs prior to fusion (Baker *et al*, 2015; Lürick *et al*, 2017, 2018; Orr *et al*, 2017). In yeast, its subunits Vps41 and Vps39 bind the Rab7-like Ypt7 protein, which is present on these organelles (Ostrowicz *et al*, 2010; Plemel *et al*, 2011; Bröcker *et al*, 2012). Curiously, Vps41 has a conserved binding motif for the AP-3 δ subunit Apl5, whose recognition requires phosphorylation of HOPS by the casein kinase Yck3 (LaGrassa & Ungermann, 2005; Cabrera *et al*, 2009, 2010).

We and others therefore postulated that the interaction of AP-3 with Vps41 occurs in the context of HOPS (Angers & Merz, 2009, 2011; Cabrera *et al*, 2010; Balderhaar & Ungermann, 2013). If this were the case, AP-3 function in membrane budding at the Golgi should proceed without a need of Vps41. Furthermore, it would imply that the coat has to stay on the vesicles at least until the tethering step, raising the question of how and when the AP-3 coat is shed off to allow for fusion. We addressed this controversy by using mitochondria as a semi-artificial tethering platform of AP-3 vesicles. By targeting Vps41 to the mitochondrial outer membrane, we show that AP-3 vesicles are specifically sequestered via the AP-3 subunit Apl5. Using the same approach, our data reveal that neither Yck3 nor Vps41 itself are required for AP-3 vesicle formation. We further identified the Arf GAP Age2 as novel factor in AP-3 trafficking. Our data suggest that uncoating of AP-3 vesicles is not required for tethering, but for their efficient fusion with vacuoles.

## Results

### Possible role of outer coat proteins on AP-3 vesicle generation

To identify components required for the generation of AP-3 vesicles, such as postulated outer coat function of HOPS Vps41, we initially localized the AP-3 subunit Apl5 relative to Vps41 and the trans-Golgi network (TGN), marked by Sec7. To block fusion of AP-3 vesicles with vacuoles after their formation, we expressed all endogenously tagged proteins in a temperature-sensitive *vam3* mutant. At the permissive temperature, mNeonGreen-tagged Vps41 was found on the vacuolar rim where it co-localized partially with the AP-3 subunit Apl5, tagged with mCherry. In contrast, Vps41-mNeonGreen was completely absent from TGN structures marked by Sec7-

mCherry, whereas Apl5-GFP co-localized with Sec7-positive structures to almost 25% (Fig 1A and B). Shifting cells to non-permissive temperature did not change co-localization between Vps41 or Apl5 and Sec7, but slightly increased co-localization of Apl5 and Vps41, indicative for an increase of tethered vesicles proximal to the vacuole.

We next asked if Vps41 is sufficient to redirect AP-3 to any membrane. We thus used an adapted anchor away approach (Haruki *et al*, 2008; Auffarth *et al*, 2014). Here, we tagged Vps41 with a C-terminal FRB-GFP tag to relocalize it to the plasma membrane, where it can bind FKBP-tagged Pma1, an abundant plasma membrane protein, upon addition of rapamycin (Auffarth *et al*, 2014). We then monitored both mCherry-tagged Apl5 and Vps41-FRB-GFP in the absence and presence of rapamycin (Fig 1C). In the absence of rapamycin, Vps41 was found exclusively on the vacuole, even though the protein was overproduced, and Apl5 was localized primarily in dots. When rapamycin was added, Vps41 was recruited strongly to the plasma membrane, and some Apl5 followed. These data suggest that Vps41 can recruit AP-3 to membranes by binding Apl5.

Previous data implied also clathrin as the main AP-3 coat (Drake *et al*, 2000). We thus monitored AP-3-dependent trafficking of a synthetic cargo to the vacuole. GFP-tagged Nyv1 carrying a C-terminal transmembrane domain of the plasma membrane SNARE Snc1, called GNS (Reggiori *et al*, 2000), is a bona fide AP-3 cargo. GNS arrives in wild-type cells at the vacuoles, but escapes to the plasma membrane before being endocytosed to some extent if the AP-3 pathway is defective (Cabrera *et al*, 2009; Fig 1D and E). In *apl5Δ* or *yck3Δ* mutants, GNS was present on the plasma membrane. However, in the absence of clathrin light chain Clc1, transport of GNS to the vacuole was not perturbed, in line with previous findings (Black & Pelham, 2000). This suggests that clathrin itself is not required for AP-3 vesicle formation, in agreement with findings in mammalian cells (Zlatic *et al*, 2011).

### Vps41 is sufficient to target AP-3 vesicles to the outer mitochondrial membrane

Having established that Vps41 alone is sufficient for AP-3 recruitment, we asked if we could use such an approach to uncover other factors involved in AP-3 vesicle formation and tethering. Mitochondria have been previously used as a platform to relocalize and analyze protein function *in vivo* (Robinson *et al*, 2010; Luo *et al*, 2014; Wong & Munro, 2014; Thomas *et al*, 2019). For us,

**Figure 1. Function of possible coat proteins in AP-3 vesicle formation.**

A   Co-localization of c-terminally GFP- or mCherry-tagged Apl5, Sec7, and Vps41 in a *vam3* temperature-sensitive (ts) background strain. Cells were grown over night at 23°C and where indicated shifted to 37°C for 3 h prior to imaging. White arrowheads indicate co-localization events. Scale bar, 5 μm. Scale bar in inset, 2 μm.

B   Quantification of co-localization events from (A). Single events were counted, and the percentage of co-localization normalized to the number of Apl5-punctae was plotted ($n \geq 187$ cells). Bars show the mean percentage of co-localization ± standard deviation. Significance was determined with a two-tailed *t*-test (***$P \leq 0.001$).

C   Re-localization of Apl5 upon Vps41 re-localization. Vps41 was tagged with a C-terminal FRBGFP tag in a strain, where Apl5 carries a C-terminal mCherry and Pma1 a FKBP-tag (Auffarth *et al*, 2014). Cells were grown in either the absence (control) or presence of 10 μM rapamycin, and localization of Vps41 and Apl5 was analyzed by fluorescence microscopy. Scale bar, 5 μm.

D   Effect of clathrin deletion on the AP-3 pathway. GFP-tagged Nyv1 with a C-terminal Snc1 transmembrane domain (GNS) was localized relative to FM4-64-stained vacuoles in the indicated strains. Labeling of the cell surface by GFP indicates a defect in the AP-3 pathway (Cabrera *et al*, 2009) Scale bar, 5 μm.

E   Quantification of AP-3 defect from (D). Linear intensity plots were laid over the cells, the AP-3 defect was calculated by dividing GFP-intensities on the vacuolar membrane by the sum of the intensity of vacuolar, and plasma membrane signal ($n \geq 30$ cells). Bars show the mean vacuolar signal over total signal ± standard deviation.

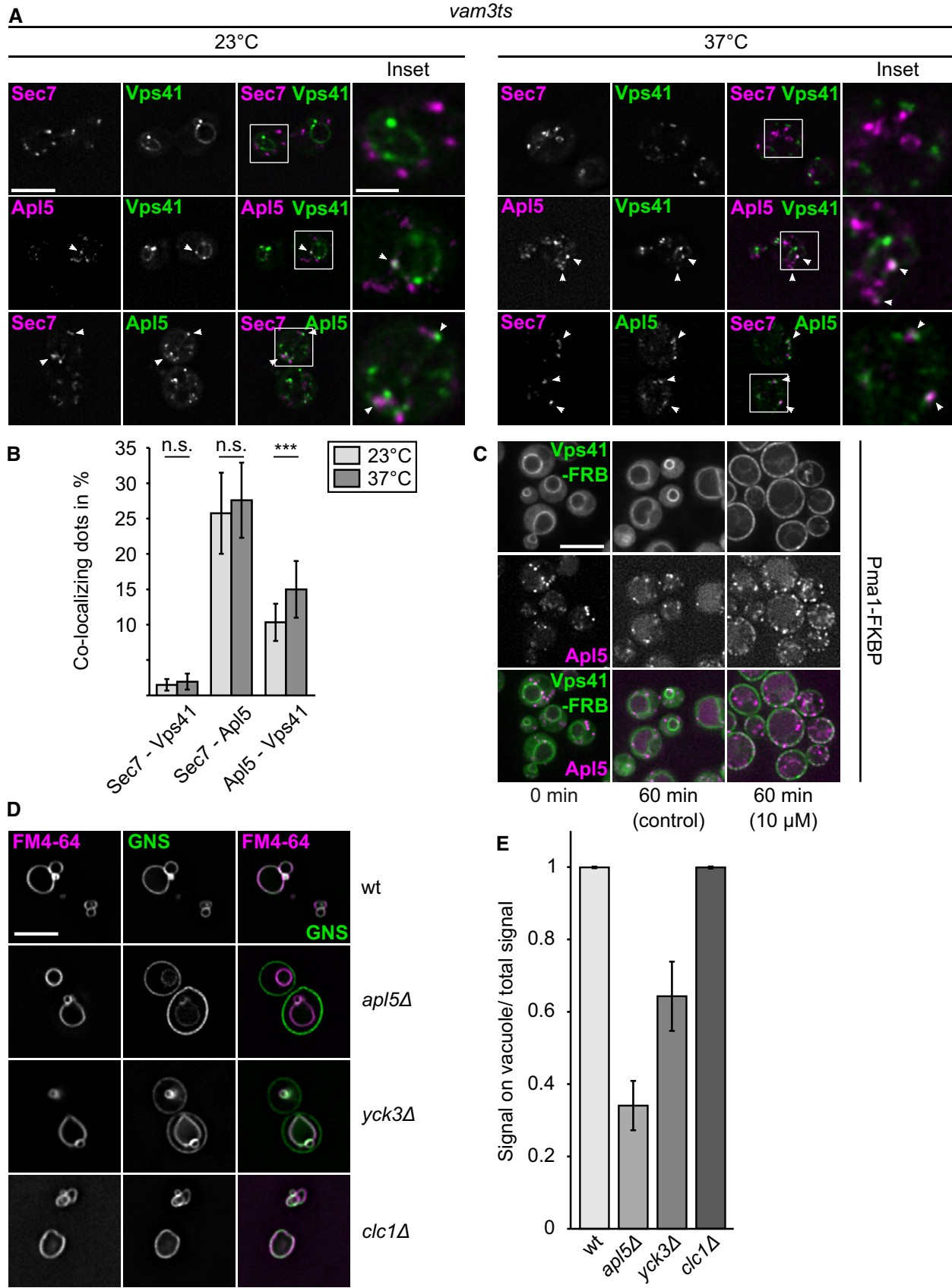

**Figure 1.**

mitochondria had the advantage that we could uncouple AP-3 vesicle tethering from their fusion, while maintaining normal HOPS function at the vacuole (Fig 2A). We therefore fused the C-terminus of Fis1, a mitochondrial localized transmembrane protein, to Vps41 and expressed the construct as an additional copy under control of the *GAL1*-promoter. If Vps41 were sufficient for tethering of AP-3 vesicles, we should see binding of these vesicles to the mitochondrial surface.

To increase the affinity of Vps41 for Apl5, we used previously established mutants of Vps41 that enhance the interaction with the AP-3 subunit Apl5, namely the SD phosphomimetic mutant (S367,368, 371, 372D), and a mutant lacking a membrane interacting amphipathic lipid packaging segment (ALPS) motif, ΔALPS. A Vps41 mutant lacking the Apl5-binding site, Vps41ΔPEST, served as a negative control (Fig 2B; Cabrera *et al*, 2010; Lürick *et al*, 2017). In all three cases, the GFP-Fis1 tagged Vps41 constructs localized efficiently to the surface of mitochondrial structures marked by DAPI staining (Fig 2C). To judge for recruitment of the assembled AP-3 complex to Vps41-marked mitochondria, we tagged β3-subunit Apl6 with mCherry, since Apl5 is the directly interacting subunit (Rehling *et al*, 1999; Darsow *et al*, 2001).

This approach indeed worked. Both the Vps41ΔALPS and the Vps41 S-D mutant accumulated numerous Apl6-positive dots (about 90%) around the mitochondria, whereas only 10% of the Apl6-positive dots co-localized with Vps41-marked mitochondria in the ΔPEST mutant (Fig 2C and D). In agreement, we co-isolated Apl5 with isolated mitochondria carrying Vps41 S-D, but not from those having the Vps41ΔPEST mutant or wild-type Vps41 (Fig 2E). Furthermore, the purified mitochondrial fractions of Vps41 S-D contained almost threefold more Nyv1, a cargo protein of the AP-3 pathway, compared to the Vps41ΔPEST mutant. As judged by the signal for Vma1, a subunit of the vacuolar ATPase, vacuole contaminations can be excluded as a sole source for this Nyv1 enrichment (Fig 2F and G). To visualize the vesicle recruitment, we turned to electron microscopy. The strains carrying the indicated mutations were grown in logarithmic phase in YPG to induce Vps41-GFP-Fis1 expression. Cells were analyzed by immuno-EM against the GFP tag. Cells expressing wild-type Vps41-GFP-Fis1 showed only some vesicular structures in close proximity to mitochondria, whereas cells expressing Vps41ΔPEST showed none at al. (Fig 3A–D). In

contrast, cells expressing the phosphomimetic Vps41 S-D mutant showed numerous small vesicles with a diameter of approximately 30–40 nm in close proximity to the mitochondrial surface (Fig 3E and F). Surprisingly, a S-A mutant of Vps41 with a functional ALPS motif showed the strongest phenotype with a high number of vesicular structures clustering between mitochondria (Fig 3G and H). This also complies with what we observed in fluorescence imaging of this strain (Appendix Fig S1A). We noticed that the GFP-staining poorly visualized mitochondrially anchored Vps41, presumably due to the shielding of the GFP epitope on the mitochondrial surface or a poor overall labeling. We thus conclude that mitochondrially bound Vps41 is sufficient to recruit AP-3 vesicles (Figs EV1 and EV2).

### Deletion of Yck3 and Vps41 does not show an effect on AP-3 vesicle generation

AP-3 vesicle generation has been controversial as proteins involved in their fusion, such as Vps41, have also been implicated in their formation (Rehling *et al*, 1999; Darsow *et al*, 2001). Using mitochondria as a specific tethering platform now allowed us to separate the role of proteins in AP-3 vesicle formation and their tethering. We first analyzed whether we could observe an AP-3 defect when we redirect the vesicles to the mitochondrial surface upon expression of the GFP-Fis1-tagged Vps41. When grown in glucose, cells expressing Vps41ΔPEST or Vps41S-D showed a clear vacuolar localization for GNS, indicating an intact AP-3 pathway. Cells grown in galactose, however, display a less clear picture of GNS. For cells expressing Vps41ΔPEST, we observed a rather minor defect in trafficking, which was more pronounced in cells expressing the Vps41 S-D mutant (Fig 4A and B). We assume that the sequestration of AP-3 to mitochondria reduces the available pool to generate AP-3 vesicles and consequently a defect in trafficking.

To determine whether previously implicated proteins involved in AP-3 vesicle trafficking are involved in vesicle formation, we deleted *VPS41* or *YCK3* in strains expressing Vps41(S-D)-GFP-Fis1, which is able to recruit vesicles to mitochondria. As a first control, we deleted the ear domains of Apl5 and Apl6 by tagging each with mCherry. Deletion of the Apl5 ear domain, which binds Vps41 (Rehling *et al*, 1999; Darsow *et al*, 2001), completely blocked its co-localization with mitochondria, whereas deletion of the Apl6 ear

**Figure 2. Mitochondrially anchored Vps41 can recruit AP-3.**

A   Model of Vps41 re-localization to mitochondria. In wild-type cells, AP-3 vesicles bud off the TGN and reach the vacuole where they are tethered by the HOPS subunit Vps41. Overexpressed Vps41, C-terminally tagged with GFP-Fis1, localizes to the mitochondria where it is able to tether AP-3 vesicles to the surface. A wild-type copy of Vps41 is still present in the background to minimize disturbance in the endogenous system.

B   Scheme of the Vps41 primary sequence with indicated motifs. The Vps41 protein harbors two motifs in its N-terminal region that have been shown to affect Apl5 binding, a PEST motif as well as an amphipathic lipid-packaging sequence (ALPS) motif (Cabrera *et al*, 2010). Furthermore, phosphorylation sites at residues S367, S368, S371, and S372 are shown, which are mutated in a phosphomimetic mutant (SD).

C   Tethering of AP-3 vesicles to the mitochondrial surface. Cells were grown at 30°C in YPG and kept in logarithmic phase. Just prior to imaging, the mitochondrial DNA was stained with DAPI as described in the methods. Localization of Apl6 and Vps41 was analyzed by fluorescence microscopy. First column shows DIC (difference interference contrast) image of cells. Scale bar, 5 μm. The inset shows the indicated magnification. Scale bar in inset, 2 μm.

D   Quantification of (C). Apl5 dots co-localizing with GFP signal were counted and plotted in percent of the total number of Apl5 dots. Bars show the mean percentage of co-localization ± standard deviation (n ≥ 60 cells).

E, F   Purification of mitochondria from selected strains. Mitochondria were purified from the indicated strains as described in the methods. The amount corresponding to 1% of the total volume was loaded as "load" sample, 5% were used for "crude" and "pure" mitochondria fractions. Samples were loaded on a 10% SDS-PAGE gel and analyzed by Western blot. White asterisks indicate unspecific bands.

G   Quantification of relative Nyv1 signal intensity from (F). Signal intensity was determined using ImageJ and a ratio of pure over crude signal intensity normalized to Tom40 signal intensity was plotted. Bars show the relative Nvy1 signal intensity ± standard deviation from *n* = 3 independent experiments.

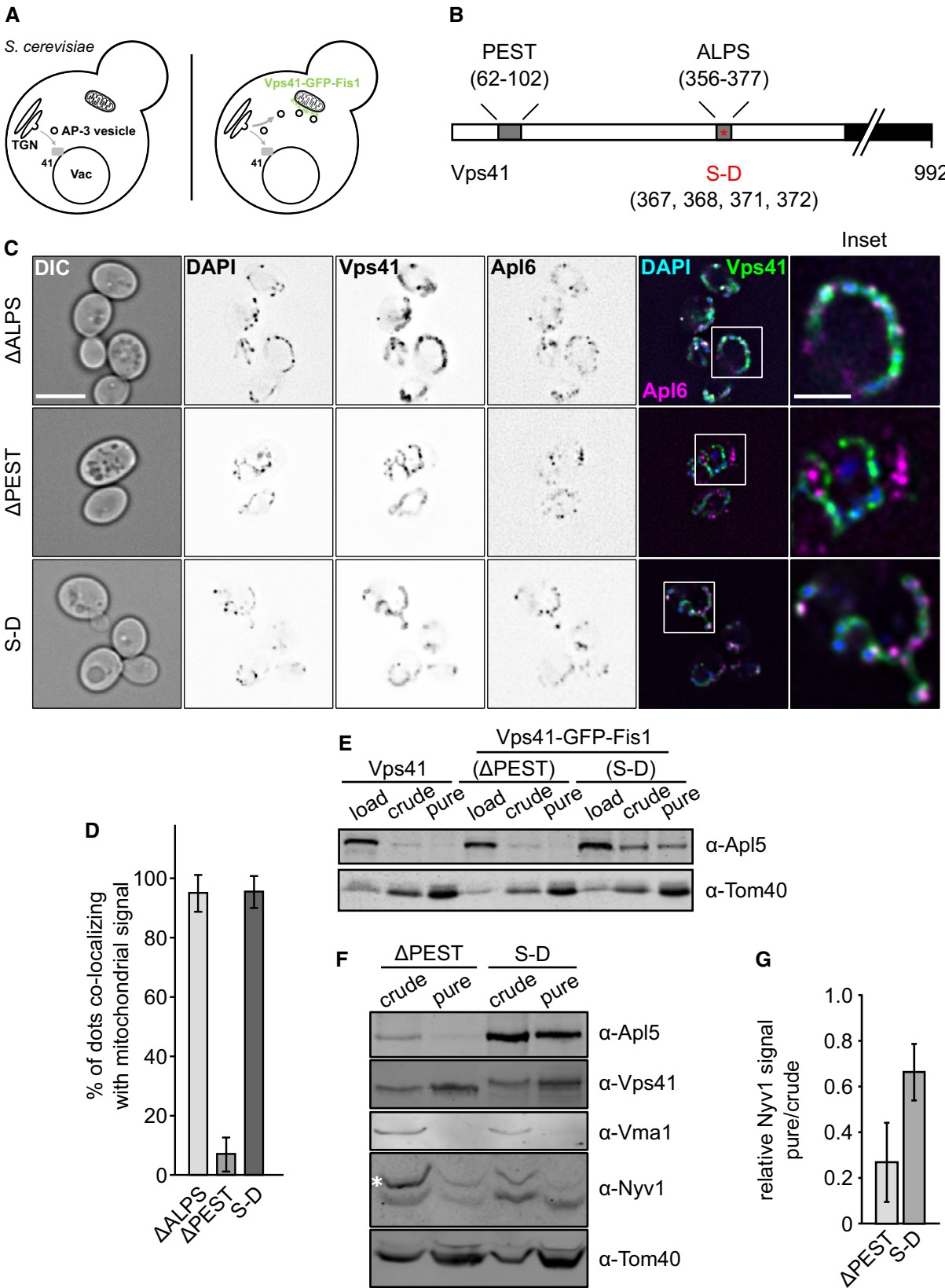

**Figure 2.**

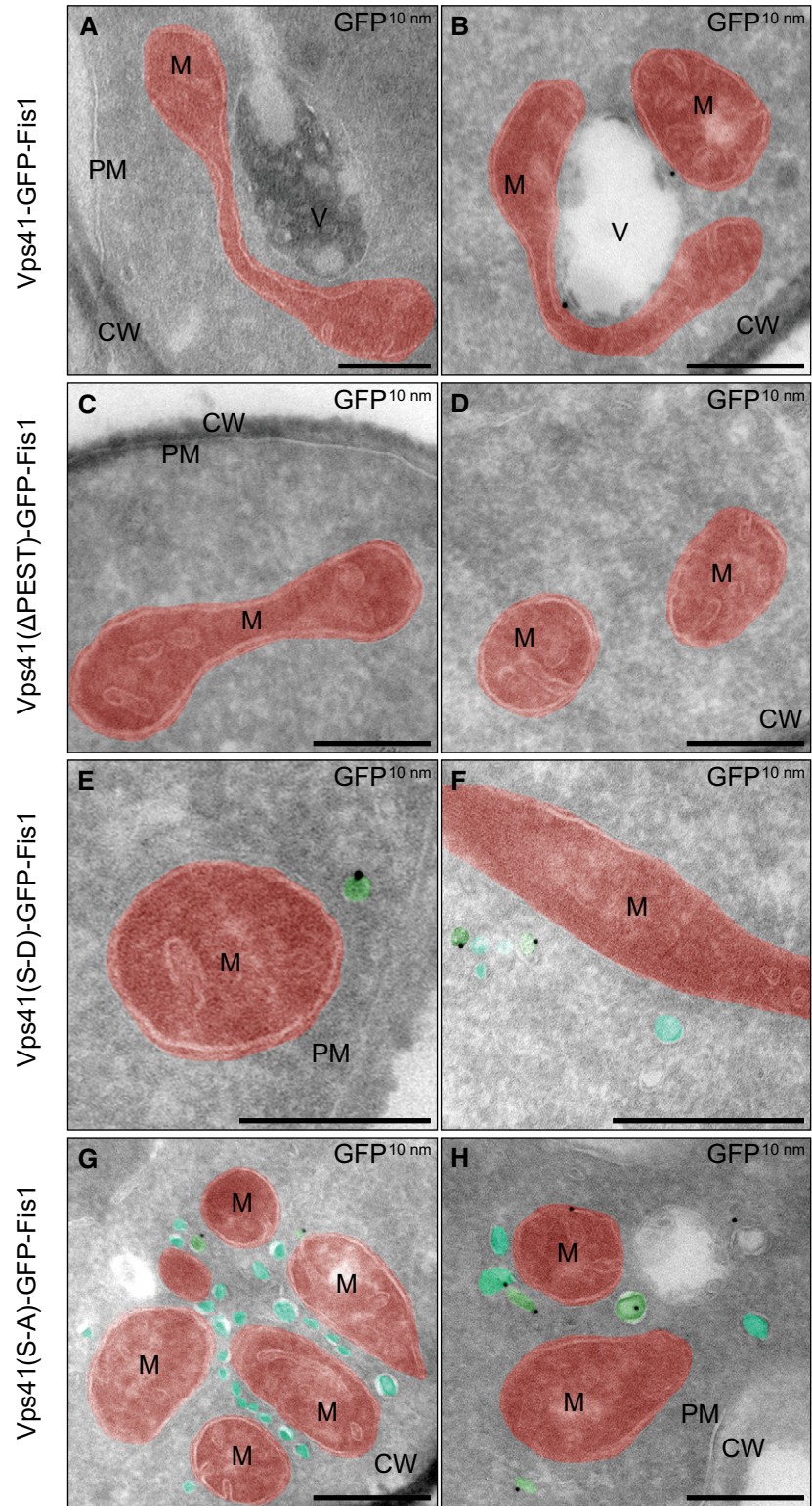

**Figure 3.  Ultrastructural analysis and immunolabeling of cells expressing mitochondrially anchored Vps41.**

A–H    Indicated strains were grown in YPG for expression of Vps41-GFP-Fis constructs. Two representative images are shown for each strain. Mitochondria (M) are
       highlighted in red and vesicular structures in teal (unlabeled for VPS41-GFP-Fis) or green (unlabeled for VPS41-GFP-Fis). Cell wall (CW), plasma membrane (PM),
       and vacuoles (V) are indicated. Scale bars, 250 nm.

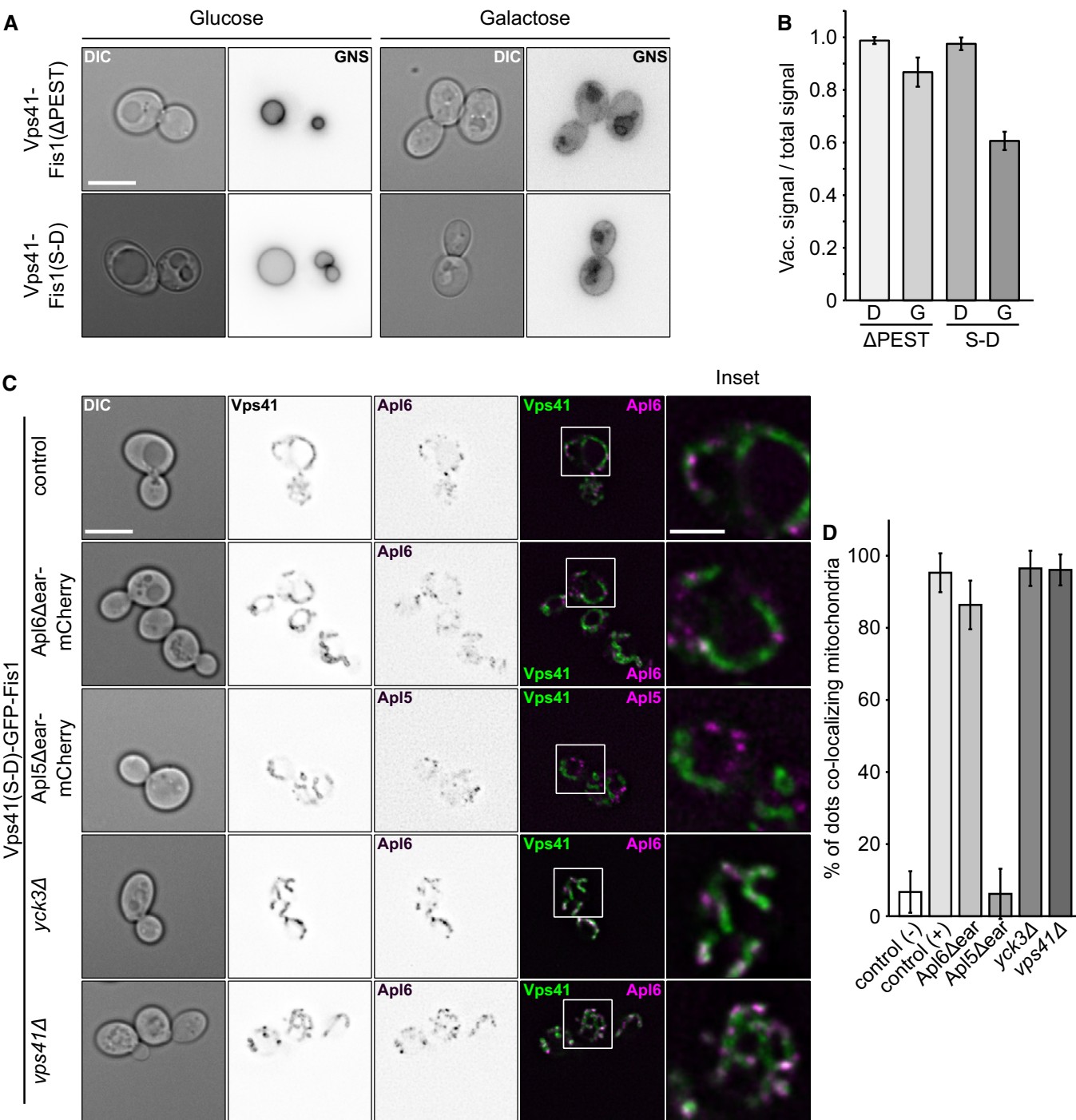

**Figure 4. Effect of mistargeting Vps41 on the AP-3 pathway.**

A GNS was localized by fluorescence microscopy in the indicated strains grown in medium containing glucose or galactose (see Fig 1D). Images were inverted to visualize possible defects. First column shows DIC image of cells. Scale bar, 5 μm.

B Quantification of the AP-3 defect from (A). Linear intensity plots were laid over cells grown in medium containing glucose (D) or galactose (G), the AP-3 defect was calculated by dividing GFP-intensities on the vacuolar membrane by the sum of the intensity of vacuolar, and plasma membrane signal (*n* ≥ 15 cells). Bars show the mean vacuolar signal over total signal ± standard deviation.

C Effect of different truncations and deletions on AP-3 targeting to mitochondria. Indicated strains were grown at 30°C in YPG and kept in logarithmic phase. Localization of Apl5 or Apl6 and Vps41 was analyzed by fluorescence microscopy. First column shows DIC image of cells. Scale bar, 5 μm. Scale bar in inset, 2 μm.

D Quantification of (C). (−) Control corresponds to Vps41(ΔPEST)-GFP-Fis1 Apl6-mCherry (CUY12558) and (+) corresponds to Vps41(S-D)-GFP-Fis1 Apl6-mCherry (CUY12560). Apl5-/Apl6-dots co-localizing with GFP signal were counted and plotted in percent of the total number of Apl5-/Apl6-dots (*n* ≥ 63). Bars show the mean percentage of co-localization ± standard deviation.

domain resulted in very efficient co-localization similar to the control (Fig 4C and D). These data revealed that localization of AP-3 to mitochondria is specific and requires the Apl5 ear domain of the AP-3 complex.

We next addressed the role of Vps41 and Yck3. Yck3-mediated Vps41 phosphorylation is required for the AP-3 pathway (Anand *et al*, 2009; Cabrera *et al*, 2009, 2010), and our data suggested that this occurs on the vacuole to make Vps41 available for tethering

(Cabrera *et al*, 2010). However, Yck3 is also transported via the AP-3 pathway to the vacuole, whereas Vps41 has been discussed as a putative AP-3 coat (Rehling *et al*, 1999; Darsow *et al*, 2001). We thus asked, where the two proteins function in the AP-3 pathway. As we localized the phosphomimetic SD mutant of Vps41 on mitochondria, which bypasses the AP-3 defect of *YCK3* deletions *in vivo* (Cabrera *et al*, 2009), we expected no influence on tethering if the proteins were not involved in AP-3 vesicle formation. Indeed,

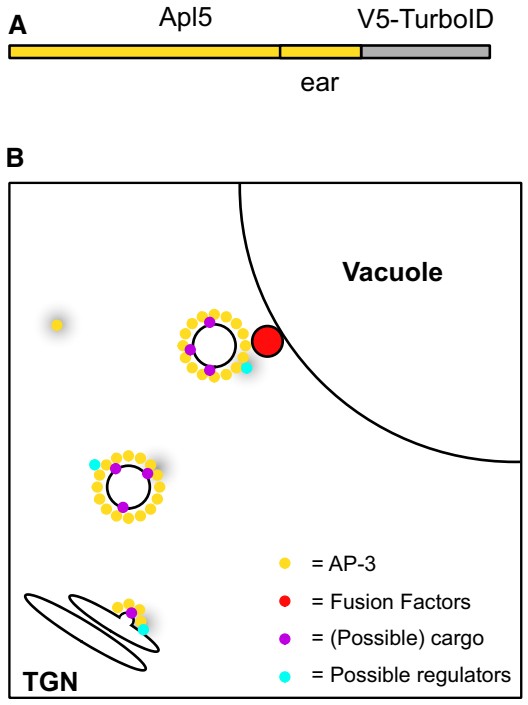

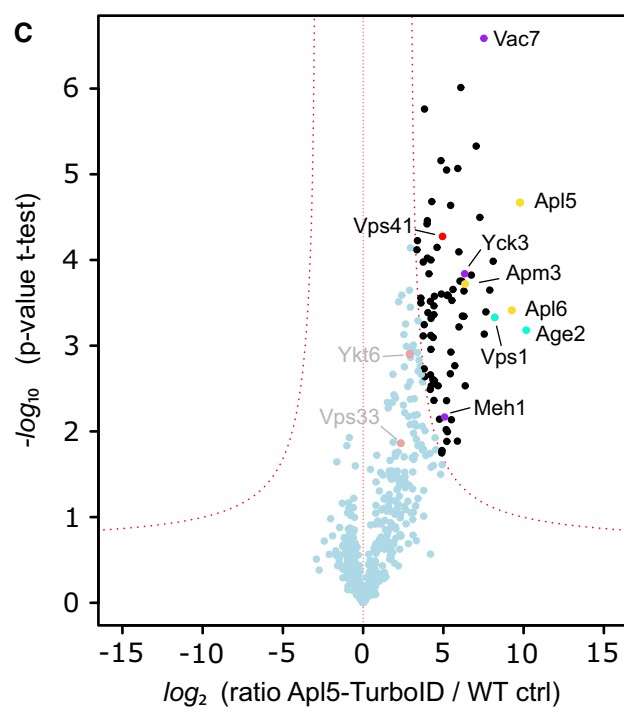

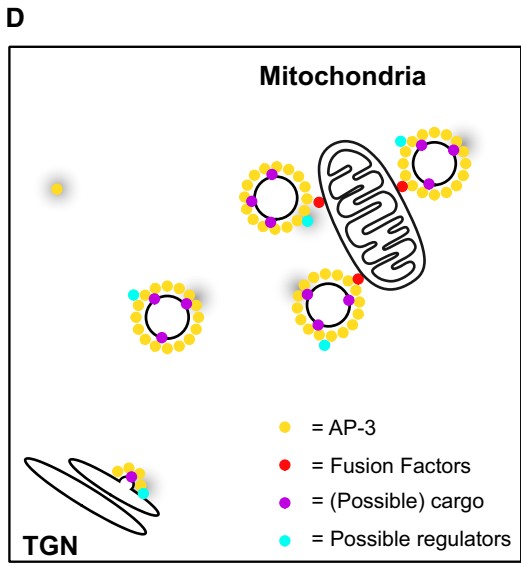

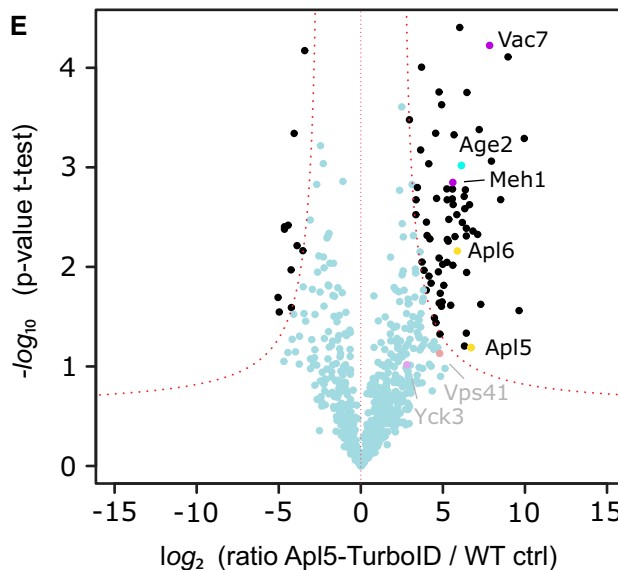

**Figure 5.**

**Figure 5.  Analysis of AP-3 interactors by proximity labeling.**

A   Scheme of Apl5 with C-terminal TurboID tag.

B   Model of possible biotinylation events along the AP-3 pathway. AP-3 has to be recruited to the TGN where it concentrates cargo and forms a vesicle. This is transported to the vacuole where it is tethered and subsequently undergoes fusion. During these steps, several interactors can come in close proximity to the TurboID Biotin Ligase (gray shade).

C   Volcano plots of proteins in close proximity to Apl5-TurboID versus control cells, from a label-free proteomics analysis of streptavidin-biotin pulldowns. The logarithmic ratios of protein intensities are plotted against negative logarithmic *P* values of two-tailed Student's *t*-test, equal variance, performed from *n* = 3 independent experiments. The red dashed line (significance, 0.05) separates specifically identified proteins (top right portion of plot) from background. Selected top hits are indicated with black dots, and all specific interactors are reported in Dataset EV1. Named hits are marked with color according to legend in (B). Gray text indicates selected non-significant hits.

D   Model of possible biotinylation events with AP-3 vesicles tethered to the mitochondrial surface by mitochondrially targeted Vps41.

E   Volcano plots of proteins in close proximity to Apl5-TurboID versus control cells in a GAL1pr-Vps41-GFP-Fis1 background, from a label-free proteomics analysis of streptavidin-biotin pulldowns with cells grown in YPG. The logarithmic ratios of protein intensities are plotted against negative logarithmic *P* values of two-tailed Student's *t*-test, equal variance, performed from *n* = 3 independent experiments. The red dashed line (significance, 0.001) separates specifically identified proteins (top right portion of plot) from background. Selected top hits are indicated with black dots, and all specific interactors are reported in Dataset EV2. Named hits are marked with color according to legend in (B). Gray text indicates selected non-significant hits.

neither deletion of *YCK3* or *VPS41* affected recruitment of Apl6-mCherry to mitochondria (Fig 4C and D). These observations agree with our interpretation that Yck3 and Vps41 are dispensable for AP-3 vesicle formation, but function in tethering AP-3 vesicles at the vacuole.

**Proximity-biotinylation reveals possible regulators of AP-3 trafficking**

Previous studies identified selected cargo proteins and regulators based on genetic screens. Here, we searched for possible regulators involved in AP-3 vesicle formation and thus tagged Apl5 C terminally with an optimized yeast Biotin Ligase (TurboID) carrying also a V5-tag to identify tagged proteins. As in other BioID approaches, the Apl5-linked enzyme should biotinylate proximal proteins in the vicinity of some 10 nm upon addition of biotin to the media (Kim *et al*, 2014; Branon *et al*, 2018).

We tagged Apl5 in the wild-type background to monitor several steps of AP-3 vesicle formation and trafficking (Fig 5A and B). The respective cells were labeled for 3 h with Biotin, and biotinylated proteins were subsequently enriched on streptavidin beads before being identified by mass spectrometry (see methods). We then plotted peptides enriched from wild-type cells relative to those expressing Apl5-TurboID and indeed found significantly more biotinylated peptides in the latter strain (Fig 5C).

First, we identified three out of four AP-3 subunits. Just the small subunit Aps3 was not detected, which may not be available to the Apl5-attached biotin ligase. Secondly, Yck3 and Meh1 (Ego1) were found as known AP-3 cargo, as well as Vac7, a subunit of the Fab1 phosphatidylinositol-3-phosphate 5-kinase (Bonangelino *et al*, 1997; Gary *et al*, 2002) (Hatakeyama *et al*, 2019), which might also traffic via the AP-3 pathway, a topic of future studies. A further interesting hit was the dynamin-like protein Vps1, which is possibly involved in AP-3 vesicle formation at the TGN.

Importantly, we only identified the HOPS subunits Vps41, the known AP-3 interactor at vacuoles, and Vps33 which, however, was not significantly enriched (shown in gray). Both localize to the same side of the HOPS complex and might thus be tagged preferentially if coated AP-3 vesicles are tethered to vacuoles (Bröcker *et al*, 2012; Balderhaar & Ungermann, 2013). Other HOPS subunits were not detected, likely due to the distance of labeling that can be achieved via BioID using Apl5. Likewise, we identified the SNARE Ykt6 non-

significantly enriched, which could be an AP-3 cargo and is involved in the fusion of AP-3 vesicles with vacuoles (Dilcher *et al*, 2001; Kweon *et al*, 2003).

A previous study identified AP-3-positive structures on vacuoles, suggesting that AP-3 vesicles might retain their coat until the tethering stage (Angers & Merz, 2009). It is known that Arf1 is required for efficient vesicle generation at the TGN, and it is likely that the small GTPase stays on the vesicle as long as it is in a GTP-bound state. This in mind, we noticed in particular one hit—Age2, an Arf1 GAP protein (Poon *et al*, 2001; Yanagisawa *et al*, 2002). Interestingly, Age2 has approximately 30% sequence identity with the AGAP1 protein which regulates AP-3 trafficking in mammalian cells (Nie *et al*, 2003).

We then used mitochondrially targeted Vps41 to enrich AP-3 vesicles and repeated the labeling via Apl5-TurboID (Fig 5D and E). Using the same protocol, we were able to identify Apl5 and Apl6, two subunits of the AP-3 complex as well as Meh1 (Ego1), a known cargo protein of the AP-3 pathway and Vac7, a potential cargo as stated above. Yck3 could be identified but not significantly enriched (Sun *et al*, 2004; Hatakeyama *et al*, 2019). Among the HOPS subunits, only Vps41 as the mitochondrial tether was identified but not significantly enriched. Importantly, we again detected Age2, suggesting that this GAP protein is present on AP-3 vesicles that were tethered to mitochondria. We thus decided to explore its role in more detail.

**Age2 is a potential uncoating factor in the AP-3 pathway**

GAP proteins conduct different functions in adaptor-mediated vesicle formation, ranging from cargo enrichment in COPII trafficking (Tabata *et al*, 2009) to vesicle uncoating in COPI trafficking via Gcs1 and Glo3 (Arakel *et al*, 2019). Age2 is one of four GAPs of Arf1 present at the Golgi and functions at the TGN (Fig 6A) (Poon *et al*, 2001). When combined with a temperature-sensitive allele of Gcs1, *age2Δ* cells show a defect in the AP-3 pathway (Poon *et al*, 2001). However, its precise role remained unclear.

We thus asked if deletion of any of these Arf GAPs alone causes a deficiency in AP-3 vesicle trafficking as analyzed by the GNS localization (Fig 1D). Only *age2Δ* cells, but not *gcs1Δ* cells, showed the GNS signal at the plasma membrane, indicative of an AP-3 defect (Fig 6B). The defect was not as severe as observed for *apl5Δ* cells,

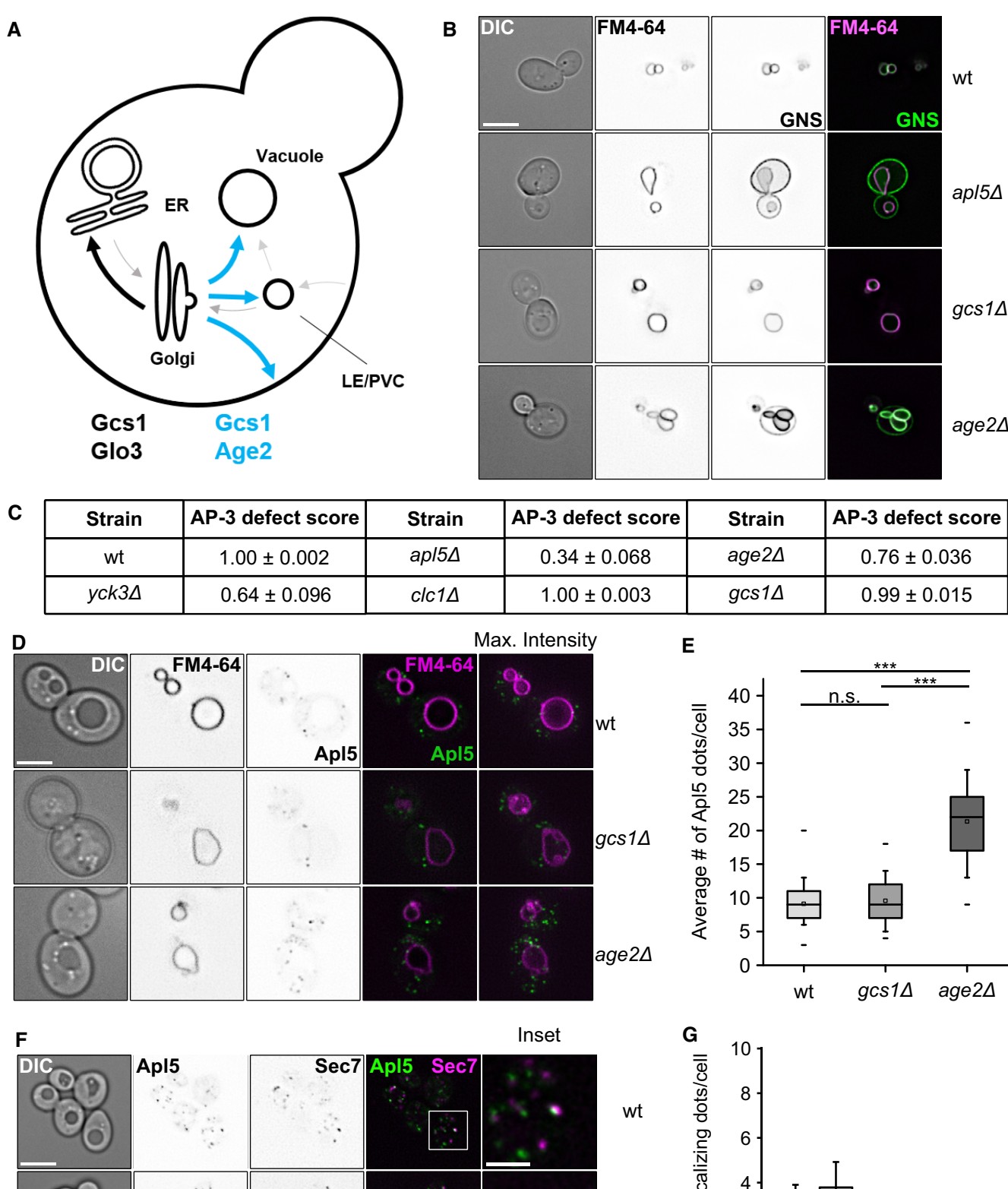

**Figure 6.  Effect of ArfGAP deletions on the AP-3 pathway.**

A   Model of ArfGAP function in vesicle trafficking through the Golgi in yeast. Gcs1 and Glo3 function in trafficking from the cis-Golgi to the ER, whereas Age2 and Gcs1 may function at or beyond the TGN (see text).

B   AP-3 sorting defects in GAP deletions. The AP-3 reporter GNS (see Fig 1D) was localized relative to FM4-64 stained vacuoles in the indicated strains. First column shows the DIC image of cells. Scale bar, 5 μm.

C   Table summarizing AP-3 trafficking defects in selected mutants.

D   Localization of AP-3 in ArfGAP deletion strains. Localization of C-terminally GFP-tagged Apl5 relative to FM4-64 stained vacuoles in the indicated strains. First column shows DIC image of cells. Scale bar, 5 μm.

E   Quantification of Apl5-dots in (D). Apl5-positive dots were counted and averaged to the number of dots per cell. Box boundaries indicate 25 and 75% of the dataset. The middle line indicates the median and the small square the mean. Whiskers correspond to 5 and 95% values, whereas the small lines show the farthermost outliers ($n \geq 110$ cells). Significance was determined with a two-tailed $t$-test (***$P \leq 0.001$).

F   Co-localization of C-terminally GFP-tagged Apl5 and C-terminally mCherry-tagged Sec7 in wt and *age2Δ* background. Cells were grown at 30°C and kept in logarithmic phase. Scale bar, 5 μm. Scale bar in inset, 2 μm.

G   Quantification of co-localization from (F) normalized per cell. Number of co-localizing dots was determined and normalized to the number of co-localization events per cell. Bars show the mean number of co-localizing dots per cell ± standard deviation ($n \geq 60$).

but comparable to deletion phenotypes of Yck3, a known regulator of AP-3 trafficking (Anand *et al*, 2009; Cabrera *et al*, 2009) (Fig 6C).

If Age2 is required for uncoating and fusion, we would expect more AP-3 positive structures in the deletion mutant, and therefore followed AP-3 by tagging Apl5 with mGFP. Indeed, the number of Apl5-positive dots increased strongly in *age2Δ* while it remains the same as in wild-type for *gcs1Δ* (Fig 6D and E).

However, we were not able to clarify whether the increase in Apl5-punctae was due to an impaired budding at the TGN or due to more vesicles in the cytosol. To answer this question, we co-localized the Apl5 dots relative to the TGN marker Sec7 in wt and *age2Δ* cells and observed similar co-localizing dots in both cells (Fig 6F and G). This suggests that the additional dots observed in *age2Δ* cells are indeed AP-3 vesicles that accumulated due to a fusion defect. However, all Apl5-positive dots found in *age2Δ* cells were efficiently recruited to mitochondrially anchored Vps41 (Appendix Fig S2A and B). In fact, the number of vesicles present on mitochondria in *age2Δ* is comparable to that of the *vps41Δ* strain, indicating that a comparable mechanism underlies this increase in numbers (Appendix Fig S2C).

### Age2 GAP activity is required for its function in AP-3 trafficking

The four known Arf1 GAPs in yeast show a high sequence similarity in their N-terminal catalytic GAP domain, which contains the conserved Arginine finger. Although the C-terminal part of each yeast ArfGAP is quite diverse, they can partially replace each other (Poon *et al*, 1999, 2001; Zhang *et al*, 2003). We therefore analyzed both Age2 and Gcs1 for their role in AP-3 vesicle trafficking.

The increase in the number of AP-3 vesicles in the *age2* deletion led us to postulate that Age2 is required for proper uncoating of the vesicles, as this was shown for other ArfGAPs before (Arakel *et al*, 2019). We thus overexpressed Age2 or Gcs1 using a TEF-promoter to test for possible premature uncoating and thus a defect in AP-3 trafficking using the GNS construct. However, in contrast to a deletion of *AGE2*, overexpression of Gcs1 or Age2 did not affect the AP-3 pathway (Fig 7A and B). We also did not detect a difference in the abundance of AP-3-positive dots (Fig 7C and D).

To ask whether the GAP activity of Age2 or Gcs1 is responsible, we generated GAP-dead mutants of Gcs1 (R54K) and Age2 (R52K). As shown above, *gcs1* deletions were without AP defect, and

expression of the GAP-dead mutant did not impair AP-3 trafficking either (Fig 7E). This suggests that Gcs1 has no exclusive role in the AP-3 pathway that other ArfGAPs cannot substitute for. In contrast, *age2Δ* cells are impaired in the AP-3 pathway (Fig 6B and D), which can be rescued by complementation with wild-type Age2, but not by the GAP-dead R52K mutant of Age2 (Fig 7F). In agreement, we observed more Apl5-positive dots only in *age2Δ* or mutant cells expressing Age2 R52K (Fig 7G and H). We thus conclude that the Age2 GAP activity on Arf1 is required for a functional AP-3 pathway. As we observe an increase of dots positive for Apl5 that do not localize with the TGN marker, we postulate that Age2 initiates uncoating of AP-3 vesicles by activating the GTPase activity of Arf1, which subsequently destabilizes the coat on the vesicular surface (Fig 8).

## Discussion

Within this study, we explored mitochondria as a tethering platform to redirect AP-3 vesicles to mitochondria. This allowed us to uncover the crucial role of the HOPS subunit Vps41 as the interactor of AP-3 vesicles and exclude a role of Vps41 as a putative coat (Rehling *et al*, 1999; Darsow *et al*, 2001; Pols *et al*, 2013). By probing the direct AP-3 environment by proximity-labeling approach, we identified the ArfGAP protein Age2 as a novel factor required for AP-3 tethering and fusion. We thus postulate that Age2 is a factor required for proper uncoating of AP-3 vesicles to allow productive initiation of the fusion process.

AP-3 vesicles transport only a selected number of proteins to the lysosome and vacuole, yet loss of its function strongly disturbs biogenesis of endosomes and vacuoles, and results in several human diseases (Dell'Angelica, 2009; Angers & Merz, 2011). In yeast, AP-3 vesicles bring the vacuolar $Q_a$-SNARE Vam3 and the R-SNARE Nyv1 to the vacuole, therefore bypassing the endosome (Darsow *et al*, 1998; Wen *et al*, 2006). The AP-3 pathway thus provides an important diversion to give vacuoles the identity of the terminal organelle of the endolysosomal system. Likewise, the SNARE VAMP7 is brought to lysosomes via the AP-3 pathway in human cells (Pols *et al*, 2013). The AP-3 pathway may thus make sure that late endosomes and vacuoles do not have the same set of SNAREs and undergo homotypic fusion before late endosomes have matured (Stepp *et al*, 1997).

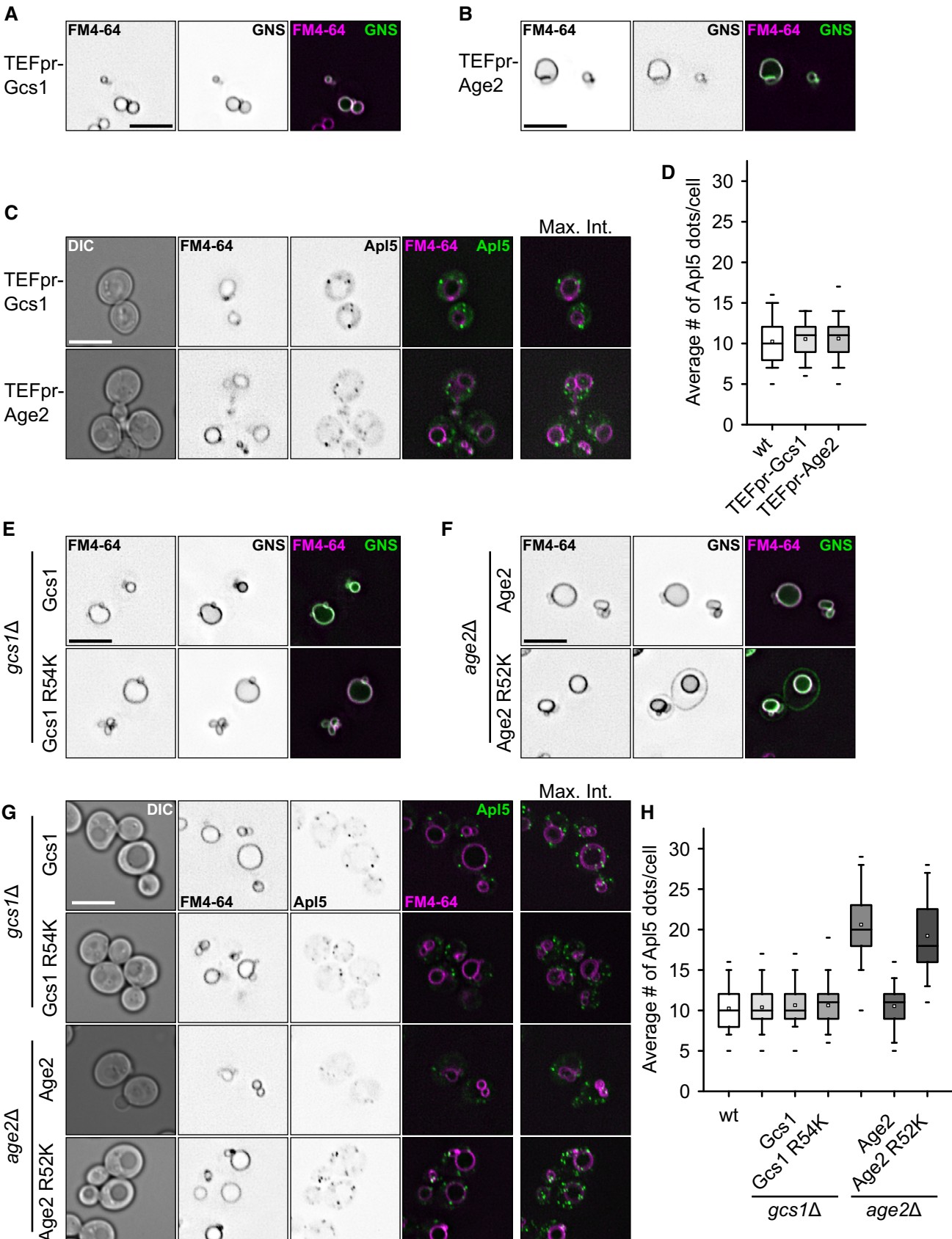

**Figure 7.**

**Figure 7. Requirement of Age2 ArfGAP activity in AP-3 trafficking.**

A, B AP-3 sorting defects in strains overexpressing Gcs1 (A) or Age2 (B). The AP-3 reporter GNS (see Fig 1D) was localized relative to FM4-64 stained vacuoles in the indicated strains. Scale bar, 5 μm.

C Localization of AP-3 in ArfGAP overexpression strains. Localization of C-terminally GFP-tagged Apl5 relative to FM4-64 stained vacuoles in the indicated strains. Scale bar, 5 μm.

D Quantification of Apl5 dots in (C). Apl5-positive dots were counted and averaged to the number of dots per cell. Box boundaries indicate 25 and 75% of the dataset. The middle line indicates the median and the small square the mean. Whiskers correspond to 5 and 95% values, whereas the small lines show the farthermost outliers ($n \geq 80$ cells).

E, F AP-3 sorting defects in GAP-dead mutants of Gcs1 (E) and Age2 (F). The AP-3 reporter GNS (see Fig 1D) was localized relative to FM4-64 stained vacuoles in the indicated strains. Scale bar, 5 μm.

G Localization of AP-3 in GAP-dead mutants of Gcs1 and Age2. Localization of C-terminally GFP-tagged Apl5 relative to FM4-64-stained vacuoles in the indicated strains. Scale bar, 5 μm.

H Quantification of Apl5 dots in (G). Apl5-positive dots were counted and averaged to the number of dots per cell. Box boundaries indicate 25 and 75% of the dataset. The middle line indicates the median and the small square the mean. Whiskers correspond to 5 and 95% values, whereas the small lines show the farthermost outliers ($n \geq 80$ cells).

Is AP-3 a special adapter protein complex that can function without an outer coat? Previous work on AP-3 suggested crosstalk with clathrin or the HOPS subunit Vps41 (Rehling *et al*, 1999; Darsow *et al*, 2001; Asensio *et al*, 2010, 2013; Pols *et al*, 2013). Indeed, AP-3 interacts via its δ-subunit Apl5 with Vps41 (Rehling *et al*, 1999; Darsow *et al*, 2001), though this interaction seems to occur rather at the vacuole than at the Golgi (Angers & Merz, 2009). By using re-localization approaches, we indeed show that Vps41 can redirect Apl5 and thus AP-3 vesicles *in vivo*, but we again find Vps41 only on the vacuole (Fig 1B and C). Likewise, redirecting AP-3 to mitochondria occurs also if *VPS41* has been deleted, indicating that Vps41 is not required to make AP-3 vesicles. Furthermore, neither in mammalian cells nor in yeast a role for clathrin in AP-3 vesicle formation (Drake *et al*, 2000) could be confirmed (Black & Pelham, 2000; Zlatic *et al*, 2013). However, exogenously overexpressed Vps41 was detected on AP-3 vesicles in human cells, which was in favor of a possible coat function implied also for other secretory organelles (Asensio *et al*, 2013; Pols *et al*, 2013). As the AP-3-binding site in Vps41 is conserved across species, we find it likely

that Vps41 interacted with AP-3 vesicles after their formation, in line with our interpretation of Vps41 as the primary vacuolar interactor as part of HOPS. Finally, as none of the so far suggested proteins, Vps41 or clathrin, seem to form an outer coat of AP-3 vesicles (and we did not find evidence in our mass spectrometric analyses either), it remains unresolved whether it is even required for AP-3 vesicles. It is possible that the cooperation of Arf1 and AP-3 is sufficient to induce high curvature in membranes and thus shape the vesicle (Ooi *et al*, 1998; Beck *et al*, 2008).

We here identified Age2 as an AP-3-specific Arf1 GAP and could separate it from the second candidate Arf1 GAP Gcs1 (Poon *et al*, 2001). Age2 is similar to human AGAP1, which has been implied in AP-3 regulation in metazoan cells, though its precise function remained unresolved (Nie *et al*, 2003). We show here that *age2Δ* cells have an AP-3 transport defect and more AP-3 positive dots in cells, which can still tether with mitochondrial Vps41 (Fig 6), and that this function of Age2 is dependent on its GAP activity towards Arf1 (Fig 7). Several Arf GAP proteins are required for uncoating of COPI and COPII vesicles (Tanigawa *et al*, 1993; Antonny *et al*, 2001; Bigay *et al*, 2003),

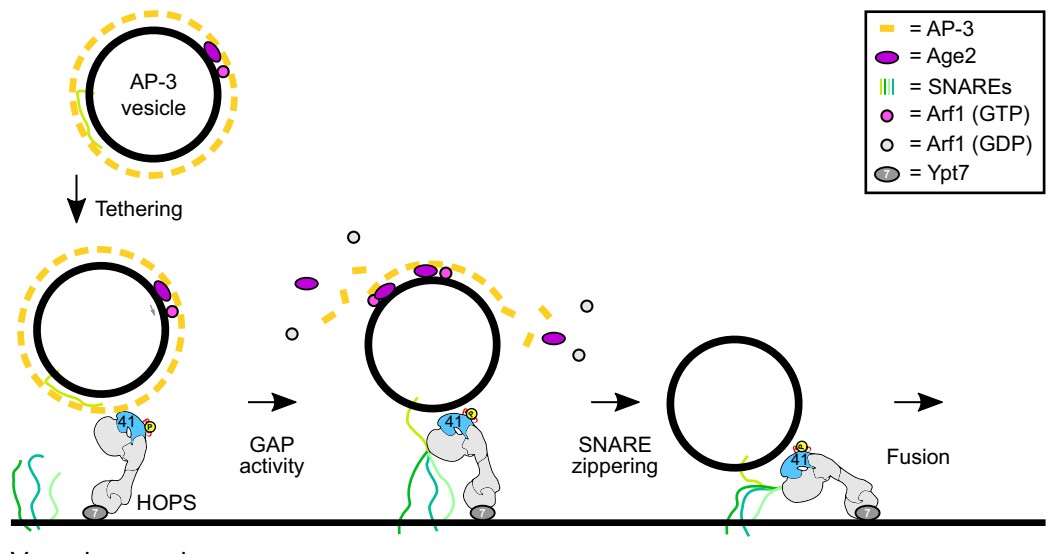

**Figure 8. Model of AP-3 vesicle tethering and fusion at the vacuole.**

AP-3 vesicles require an assembled coat to interact with vacuole-localized HOPS, and bind to Vps41. Interaction with vesicle SNAREs capture AP-3 vesicles stably and allow partial uncoating via the Arf GAP Age2. SNARE and HOPS interactions then drive fusion with the vacuole.

yet it remains controversial if uncoating generally occurs immediately after vesicle budding or upon tethering (Angers & Merz, 2011). In this regard, the interaction of the ER-localized Dsl1 tethering complex with the COPI coat (Andag & Schmitt, 2003; Ren et al, 2009; Meiringer et al, 2011) is intriguing as it suggests a crosstalk of a coated vesicle with a tether (Angers & Merz, 2011).

Recent work on COPI revealed a close interplay of two ArfGAPs, Glo3, and Gcs1 in uncoating, one of which is controlled in activity by phosphorylation (Arakel et al, 2019). COPI binds two Arf1 proteins in different molecular environments (Bykov et al, 2017; Dodonova et al, 2017), as does AP-1 (Shen et al, 2015), suggesting that one Arf GAP protein may destabilize the coat partially. Age2-mediated Arf1 hydrolysis and final uncoating may thus occur only very late, possibly after tethering of AP-3 vesicles at the vacuole and engagement of AP-3 localized SNAREs with the vacuolar HOPS complex, which has several SNARE-binding sites (Angers & Merz, 2011; Lürick et al, 2015; Orr et al, 2017). This would agree with the observation of vacuole-localized AP-3 subunits and the crosstalk of Vps41 with Apl5. Future studies will need to clarify, when and how Age2 functions in AP-3 trafficking.

It has been assumed that coats are rather instable once vesicles have formed. Our data rather suggest that vesicles retain their coat as part of their identity, and productive and complete shedding may be tightly linked to fusion. Arf GAPs may play a critical role here. They function in cargo concentration at budding sites of vesicles as shown for Sar1 (Futai et al, 2004; Tabata et al, 2009) and Gcs1 (Arakel et al, 2019), and promote uncoating, likely in coordination with the tethering and fusion machinery, as suggested for Age2 here. How the coat and uncoating machinery as well as fusion machinery cooperate and regulate each other's activity is an exciting and rather unexpected avenue to be explored in the trafficking field.

# Materials and Methods

### Yeast genetic manipulation and molecular biology

S. cerevisiae strains used are listed in Table S1. Genetic manipulations were made by homologous recombination of PCR fragments as described previously. A plasmid encoding GFP-Nyv1-Snc1 (Reggiori et al, 2000) was transformed into the indicated strains. Plasmid pRS415-TurboID-V5 was a gift from Alice Ting (Addgene plasmid #107167; http://n2t.net/addgene:107167; RRID:Addgene_107167) (Branon et al, 2018). The TurboID-V5 tag was cloned into a pYM30 vector for subsequent integrative tagging. Plasmids encoding Age2 and Gcs1 constructs including respective promoter and terminator regions were generated by cloning of the coding sequence (± 500 bp) into pRS406 background vector (pRS406-AGE2pr-AGE2-AGE2term.; pRS406-GCS1pr-GCS1-GCS1term.). Mutations were introduced by Q5® Site-directed Mutagenesis (New England Biolabs) (pRS406-AGE2pr-AGE2(R52K)-AGE2term.; pRS406-GCS1pr-GCS1(R54K)-GCS1term.). See table S1 for overview of yeast strains.

### Light microscopy and image analysis

Cells were grown to log-phase in yeast extract peptone (YP) medium containing glucose (YPD), galactose (YPG), or synthetic medium supplemented with essential amino acids (SDC). The vacuole membrane was stained with 30 μM FM4-64 for 30 min, followed by washing with medium, and incubation in medium without dye for 1 h before analysis. For staining of mitochondrial DNA, cells were incubated in 2 μg/ml 4′,6-Diamidine-2′-phenylindole (DAPI) for 20 min in the dark and washed with SDC. Images were acquired on an Olympus IX-71 inverted microscope equipped with a 100× NA 1.49 objective, a sCMOS camera (PCO), an InsightSSI illumination system and SOft-WoRx software (Applied Precision). Images were processed with ImageJ. One representative plane of a z-stack is shown unless noted.

### Purification of mitochondria

For purification of yeast mitochondria, 2 l of YPG were grown to an $OD_{600} = 1$. Cells were harvested and washed once with ddH$_2$O, and mitochondria were isolated as described previously (Zahedi et al, 2006; Montoro et al, 2018) with minor alterations. In brief, cells were resuspended in DTT buffer (0.1 M Tris-HCl, pH 9.4, 10 mM DTT) at 30°C for 15 min. Pellets were resuspended in spheroplasting buffer (0.2 × YPD, 0.6 M sorbitol, 50 mM potassium phosphate, pH 7.4) and incubated at 30°C for 30 min with occasional inverting. Spheroplasts were resuspended in breaking buffer (0.6 M sorbitol, 10 mM Tris-HCl, pH 7.4, 1 mM EDTA, 0.2% [w/v] BSA [essentially fatty acid-free, Sigma-Aldrich], 1 mM PMSF, 0.05 × PIC), and homogenized with 12 strokes in a tight-fitting glass potter. Mitochondria were enriched by differential centrifugation, and purity was further increased by separation on a sucrose gradient. Following two centrifugation steps at 1,500 g and 4,000 g, mitochondria were enriched in the last pellet fraction at 12,000 g. Crude mitochondria were resuspended in SEM buffer (250 mM sucrose, 1 mM EDTA, 10 mM MOPS-KOH pH, 7.2), and the protein concentration was determined. The mitochondrial suspension was diluted to a concentration of 5 mg/ml, and 1 ml was loaded on top of a 4-step sucrose gradient consisting of 60, 32, 23 and 15% (w/v) of sucrose in EM buffer (1 mM EDTA, 10 mM MOPS-KOH, pH 7.2). Following 90-min centrifugation at 4°C and 135,000 g in a SW40 rotor, mitochondria were collected from the 60 to 32% interface, and concentrated by centrifugation at 10,000 g. Protein concentration was determined, and equal amounts of the crude and pure fractions were analyzed by Western Blot.

### Immuno-electron microscopy

Strains were grown in YPG to exponential phase and fixed, embedded in 12% gelatin, and cryo-sectioned as previously described (Griffith et al, 2008). Ultrathin cryo-sections were immunogold labeled using a rabbit anti-GFP (Abcam, ab290), followed by a detection with protein A-gold 10 nm conjugate (Cell Microscopy Center, University Medical Center Utrecht, The Netherlands). Cell sections were imaged using a 80KV FEI-Cm100bio electron microscope equipped with a digital camera (Morada, Olympus).

### Purification of biotinylated proteins via streptavidin beads

Cells expressing Apl5-TurboID-V5 and wild-type cells were grown over night in YPD to an $OD_{600}$ of 0.4. 100 μM biotin were added, and cells were grown for another 3 h to a final $OD_{600}$ of 0.8–1. 750 $OD_{600}$ equivalent units of cells were harvested by centrifugation at 3,000 g for 10 min. Cells were washed two times with ddH$_2$O, and the pellet was resuspended in lysis buffer (2 M NaOH, 7.5% β-

mercaptoethanol) and incubated on ice for 15 min. Proteins were precipitated with 22.5% Trichloroacetic acid (TCA), and the pellet was washed three times with ice-cold acetone. Pellets were dried and incubated in resuspension buffer (4 M Urea, 0.5% SDS, 10 mM DTT) until they were completely dissolved. Magnetic streptavidin beads were added to the suspension, and tubes were incubated on a turning wheel at RT for 1.5 h. The beads were washed six times with washing buffer (0.5% SDS, 10 mM Tris-HCl, pH 8.0). Bound proteins were eluted by boiling beads in sample buffer at 99°C followed by rapid removal of the supernatants. Samples were loaded on a 10% SDS-gel for some minutes, and the top part of each lane was excised and processed by in gel-digestion with LysC as described previously (Eising *et al*, 2019).

### Mass spectrometry and analysis of biotinylated peptides

Peptides were separated by HPLC (Thermo Ultimate 3000 RSLCnano) on a 50-cm PepMap® C18 easy spray columns (Thermo) with an inner diameter of 75 μm at a constant temperature of 40°C. The column temperature was kept at 40°C. Peptides were eluted from the column with a linear gradient of acetonitrile from 10 to 35% in 0.1% formic acid for 32 min at a constant flow rate of 250 nl/min. Peptides eluting from the column were directly electrosprayed into a Q Exactive *Plus* mass spectrometer (Thermo). Mass spectra were acquired on the Q Exactive *Plus* in a data-dependent mode to automatically switch between full MS scan and up to ten data-dependent MS/MS scans. The maximum injection time for full scans was 50 ms, with a target value of 3,000,000 at a resolution of 70,000 at $m/z = 200$. The ten most intense multiply charged ions ($z = 2$) from the survey scan were selected with an isolation width of 1.6 Th and fragment with higher energy collision dissociation (Olsen *et al*, 2007) with normalized collision energies of 27. Target values for MS/MS were set at 100,000 with a maximum injection time of 120 ms at a resolution of 17,500 at $m/z = 200$. To avoid repetitive sequencing, the dynamic exclusion of sequenced peptides was set at 20 s. The resulting MS and MS/MS spectra were analyzed using MaxQuant (version 1.6.0.13, www.maxquant.org/; (Cox and Mann, 2008; Cox *et al*, 2011) as described previously (Fröhlich *et al*, 2013). The search included carbamidomethylation of cysteine as a fixed modification and methionine and lysine biotinylation ($C_{10}H_{14}N_2O_2S$; mass change 226.077598394) was added as a variable modification. The maximum allowed mass deviation was 6 ppm for MS peaks and 20 ppm for MS/MS peaks. The maximum number of missed cleavages was three, due to potential missed cleavages caused by biotinylated lysines. The false discovery rate was determined by searching a reverse database. The maximum false discovery rate was 0.01 on both the peptide and the protein level. The minimum required peptide length was six residues. All experiments were performed in triplicates and analyzed using the label-free quantification option of MaxQuant (Cox *et al*, 2014). Calculations and plots were performed with the R software package (www.r-project.org/; RRID:SCR_001905) following available label-free quantification scripts (Hubner *et al*, 2010).

## Data availability

The mass spectrometry proteomics data have been deposited to the ProteomeXchange Consortium via the PRIDE partner repository with the dataset identifier PXD020623 (https://www.ebi.ac.uk/pride/) and 10.6019/PXD020623.

**Expanded View** for this article is available online.

## Acknowledgements

We thank Kathrin Auffarth and Angela Perz for expert technical assistance. This work was supported by grants of the DFG (SFB 944, project P11 to C.U., and P20 to F.F.) and the NWO open program ALW.355 (M.M.). Open access funding enabled and organized by Projekt DEAL.

## Author contributions

JS conducted experiments, analyzed data, and wrote the initial draft, MM prepared and analyzed immuno-EM images, EY, MC, and KA conducted initial fluorescence microscopy, SW conducted MS analysis, FF analyzed all MS data, CU supervised the project and wrote the manuscript and final draft, which was approved by all authors.

## Conflict of interest

The authors declare that they have no conflict of interest.

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
