## [Review Process File · The EMBO Journal]

AP-3 vesicle uncoating occurs after HOPS-dependent vacuole tethering

Jannis Schoppe, Muriel Mari, Erdal Yavavli, Kathrin Auffarth, Margarita Cabrera, Stefan Walter, Florian Fröhlich and Christian Ungermann,

DOI: [10.15252/embj.2020105117](https://doi.org/10.15252/embj.2020105117)

Corresponding authors: Christian Ungermann (cu@uos.de)

Review Timeline:

Submission Date:	27th Mar 20
Editorial Decision:	20th Apr 20
Revision Received:	23rd Jun 20
Editorial Decision:	9th Jul 20
Revision Received:	16th Jul 20
Accepted:	22nd Jul 20

Editor: Elisabetta Argenzio

Transaction Report:

Thank you for submitting your manuscript entitled "AP-3 vesicles retain their coat and depend on the HOPS subunit Vps41 just for tethering to vacuoles" [EMBOJ-2020-105117] to The EMBO Journal. Your study has been sent to three reviewers for evaluation, whose reports are enclosed below.

As you can see, the referees consider the work potentially interesting. However, while referee #1 and #2 are overall positive and request you to address only minor points, reviewer #3 raises several criticisms that need to be solved before they can support publication in The EMBO Journal. In particular, this referee requests you i) to provide more direct evidence on the role of Age2 in vesicle uncoating; ii) to further investigate the relevance of GSN accumulation at the plasma membrane and its endocytosis; and iii) to test why there is not a bigger shift toward the vacuole when fusion is blocked.

In addition to the points listed above, addressing all major and minor referees' criticisms will be essential for the publication of your work in The EMBO Journal. I should also add that it is our policy to allow only a single round of major revision. Therefore, acceptance of your manuscript will depend on the completeness of your responses in this revised version.

We generally grant three months as standard revision time. As we are aware that many laboratories cannot function at full capacity owing to the COVID-19 pandemic, we may relax this deadline. Also, we have decided to apply our 'scooping protection policy' to the time span required for you to fully revise your manuscript and address the experimental issues highlighted herein. Nevertheless, please inform us as soon as a paper with related content published elsewhere.

Referee #1:

This manuscript makes an important advance in the area of vesicle coats. The adaptin complex was always seen as a layer of a coat, not as an independent coat. My interest in the manuscript is triggered by the unequivocal demonstration that AP-3 vesicles can form without HOPS or clathrin. This is a generally relevant finding adding to the fundamental question of how coats work.

Similarly, we are now in a phase where the redundant functions of different ArfGAPs at an integrated level ("secretion" per se) is mechanistically dissected. The finding that Age2 is specifically relevant to AP-3 promotes understanding of the diverse and versatile functions of ArfGAPs.

I have minor comments Figure by Figure:

1) Legend for B needs a statement of the statistical test used and should explain the meaning of the three asterisks. Conceptually, I am not so happy with the term "vesicle biogenesis" and strongly prefer "vesicle formation" or "vesicle generation". At least in the context of other coats this seems the preferred term.

2) Nice data. You make a quantitative statement based on 2F (Nyv signal). At least, the legend should specify how often the experiment was repeated to be able to make this statement.

3) The Results do not first talk about 3A and 3B. Is 3E missing? Please check description. the data on the Apl5_delta_ear is very strong.

4) In principle this is nice and strong data. The one aspect that confuses me is that no proteins of the mitochondrial outer membrane are labelled in the plot. If the mitochondrially tethered population of Apl5 is too small to give a change in the labelling pattern then I wonder whether this experiment is worth showing. In any case it should be presented better why both situations were analyzed. What was the expectation? Why are no mitochondrial proteins labelled? Why was it then decided to show the experiment?

5) Although the conclusion is that no significant change was observed, which seems plausible 5G needs an explanation in the Legend what the error bars show (SD, SEM?)

6) Is labelled Figure 7 in the Figures. What is the light pink shape "attacking" the tethered vesicle?

Referee #2:

This study from Schoppe et al. addresses a few outstanding issues in the realm of AP-3 biology. The precise role of Vps41 in AP-3 trafficking has remained unclear. There is good evidence in the literature from the Ungermann group and others for Vps41 functioning as part of the HOPS complex to tether AP-3 vesicles to the lysosome/vacuole. However, an additional role of Vps41 in biogenesis of AP-3 vesicles has been proposed. This study addresses this potential role primarily through the use of an elegant anchor-away approach in which they ectopically localize Vps41 to mitochondria. Through these experiments, the authors provide convincing evidence that Vps41 is not required for AP-3 vesicle biogenesis but is sufficient for AP-3 vesicle tethering. They also show that the kinase Yck3 is important for vesicle tethering but not biogenesis. There are nice control experiments throughout the work, such as the requirement of the Apl5-ear domain for the anchor-away interaction between Vps41 and the AP-3 complex. In addition, they used a biotinylation (BioID) approach to identify proteins interacting with (or nearby) Apl5 in both w.t. and anchor-away cells. Using this approach they identified Age2 as an unexpected interactor. Age2 is an ArfGAP and a homolog of AGAP1, which has been implicated in regulation of AP-3 in mammalian cells. The authors interpret the fact that Age2 is found to interact with Apl5 in both wt and anchor-away cells to mean that Age2 may be involved in the uncoating process. Importantly, they demonstrate that Age2 is not required for AP-3 vesicle biogenesis (in fact there are more vesicles formed in age2 mutants) and therefore Age2 is very likely to function to uncoat the AP-3 vesicle.

Overall I found this study to be very convincing and interesting. It also opens the field up for further

studies investigating the precise mechanisms of vesicle uncoating. I have only minor suggestions for improvement.

1. On page 6 the authors cite a few previous studies using mitochondrial anchor-away to test protein-protein interactions. I think it would be good to also cite the work from Wei Guo's group where they anchored the exocyst subunit Sec3 to mitochondria and observed secretory vesicle tethering (Luo et al., MBoC 2014).

2. Figure panel 3A appears to not be specifically described in the results section?

3. Circumstances permitting, it might be interesting to test whether the catalytic Arg residue of Age2 is required for Age2 function in their various AP-3 functional tests. This would provide more evidence for GAP activity as the key function of Age2 in AP-3 regulation. However, this is a relatively minor point and with research facilities shut down there is no reason to prevent publication without these experiments.

Referee #3:

This manuscript reports that the previously demonstrated interaction of the Vps41 subunit of HOPS with the Apl5 subunit of AP-3 is responsible for tethering AP-3 vesicles to the vacuole. This conclusion is in line with the function of the HOPS complex in vesicle tethering but contrasts with previous work suggesting that Vps41 functions as a clathrin-like coat for the formation of AP-3 vesicles. In addition, the authors report that the Age2 Arf1-GAP protein mediates AP-3-dependent sorting and probably uncoating of AP-3 vesicles. These conclusions are interesting, but, in my opinion, they don't rise to the level of novelty expected for a journal like EMBO J. Some of the findings presented in this paper have already been shown before, though in somewhat different ways. In addition, some of the results are difficult to appreciate, probably because of the challenge to visualize intracellular structures in yeast. Finally, there are some questions that are relevant but the manuscript leaves unanswered.

Specific points:

Are other subunits of the HOPS complex involved in tethering AP-3 vesicles to the vacuole or does Vps41 function independently of the other HOPS subunits? Vps39 KO also interferes with the AP-3 pathway. Is it also involved in tethering? Why isn't it detected in the BioID experiments? The role of HOPS and its individual subunits in tethering AP-3 vesicles is a fundamental issue that the study doesn't address.

Although KO of Age2 increases the number of AP-3 vesicles, this observation doesn't necessarily mean that it plays a role in uncoating; other scenarios are possible. More direct evidence for a role in uncoating is needed.

Fig. 1A,B. The difference in co-localization of Apl5 with Vps41 at the permissive and non-permissive temperature is statistically significant but very small (5%?). If there is a block in fusion, why isn't there a bigger shift toward the vacuole? And why doesn't the amount of Apl5 in the Sec7 compartment decrease?

Fig. 1D,E. The independence of AP-3-mediated sorting from clathrin was already shown in previous

studies, so these experiments add little to what is already known. With regards to the results, why does GNS accumulate at the plasma membrane in AP-3 mutants? Why is it not endocytosed (as mentioned in the text)? Is AP-3 involved in endocytosis?

Fig. 2C. The results are very difficult to appreciate. I suggest that they show bigger pictures of the yeast cells.

Fig. 2F. What band is Nyv1? Differences in crude and pure mitochondrial fractions of both constructs are difficult to see. The text mentions a 3-fold difference that is not clear from these images.

Figs. 3A and 3B are not described in the text. If they are not necessary, move them to a supplementary figure or delete.

Fig. 3C. Make it larger so that differences can be better appreciated.

Fig. 5F. The additional number of Apl5 dots in Age2 mutants is not clear in this panel.

Use "Inset" in place of "inlet" in the figures.

We thank all reviewers for their constructive feedback. The following changes were done in the revised manuscript:

- Figure 2F was repeated and the added quantification (Figure 2G) shows now more convincingly that AP-3 cargo is captured on mitochondria carrying Vps41
- We added a new figure 3, where we show that mitochondrially anchored Vps41 recruits vesicles to mitochondria. This phenotype is observed for the Vps41-SD mutant, but not at all for the Δ PEST mutant that lacks the AP-3 binding motif. Surprisingly, we observed far more vesicles in the Vps41-SA mutant, which we took along as a control. This mutant has a functional ALPS motif, which is able to bind to small vesicles (Cabrera, Langemeyer et al., 2010). We did not detect more AP-3 vesicles on mitochondria than in the Vps41 SD mutant (Figure S1), and thus suspect that also other vesicles are captured. We take this as an indication of a functional ALPS motif, though did not analyze this further.
- The new Figure 7 shows that Age2 GAP dead mutant, but not Gcs1 GAP dead causes an AP-3 sorting defect, whereas overexpression of the GAPs did not cause any defect. This provides further support for a direct role of Age2 in AP-3 vesicle uncoating.
- We fixed mistakes in figure references and revised the text to clarify several open issues.
- Due to the addition of the electron microscopy data and mass spectrometry, two authors were added (Muriel Mari, Stefan Walter), and all authors agreed to this.

Our detailed response is given below.

Referee #1:

This manuscript makes an important advance in the area of vesicle coats. The adaptin complex was always seen as a layer of a coat, not as an independent coat. My interest in the manuscript is triggered by the unequivocal demonstration that AP-3 vesicles can form without HOPS or clathrin. This is a generally relevant finding adding to the fundamental question of how coats work.

Similarly, we are now in a phase where the redundant functions of different ArfGAPs at an integrated level ("secretion" per se) is mechanistically dissected. The finding that Age2 is specifically relevant to AP-3 promotes understanding of the diverse and versatile functions of ArfGAPs.

We thank the reviewer for her/his positive feedback.

I have minor comments Figure by Figure:

1) Legend for B needs a statement of the statistical test used and should explain the meaning of the three asterisks. Conceptually, I am not so happy with the term "vesicle biogenesis" and strongly prefer "vesicle formation" or "vesicle generation". At least in the context of other coats this seems the preferred term.

Thank you for pointing this out. We added the required information to the Figure legend of Figure 1B and changed the term in the text in all cases.

2) Nice data. You make a quantitative statement based on 2F (Nyv signal). At least, the legend should specify how often the experiment was repeated to be able to make this statement.

Thank you for this suggestion. The number of repetitions was added to the figure legend and a graph showing the quantification was added to the Figure (Figure 2G).

3) The Results do not first talk about 3A and 3B. Is 3E missing? Please check description. the data on the Apl5_delta_ear is very strong.

Indeed, we made a mistake here. We added this in the text and changed the wrong reference for Figure 3E (which is now Figure 4E).

4) In principle this is nice and strong data. The one aspect that confuses me is that no proteins of the mitochondrial outer membrane are labelled in the plot. If the mitochondrially tethered population of Apl5 is too small to give a change in the labelling pattern then I wonder whether this experiment is worth showing. In any case it should be presented better why both situations were analyzed. What was the expectation? Why are no mitochondrial proteins labelled? Why was it then decided to show the experiment?

The main purpose of this experiment was to provide further evidence of vesicles tethered to the mitochondria and not only AP-3 complex. It is true, that the labeling pattern does not change much, rather the overall rate of detection even decreases. Considering that the radius of efficient biotin-labeling is approximately 10 nm, it is not surprising that we do not see a lot of mitochondrial proteins labeled (Kim et al., 2014). Membrane-bound Vps41 tethers the vesicles to the mitochondria and this distance would need to be bridged. In addition, we cannot say where exactly the Biotin-Ligase is located within the AP-3 complex if it is attached to the C-terminal part of Apl5. A detailed structure (including hinge and ear domains) of any AP-complex is so far missing. We believe that this can explain the lack of labeling of mitochondrial proteins. Importantly, the labeling of cargo molecules indicates that we captured complete vesicles at mitochondria.

5) Although the conclusion is that no significant change was observed, which seems plausible 5G needs an explanation in the Legend what the error bars show (SD, SEM?)

Thank you for pointing this out, we added the specifications in the Figure legend to Figure 6G (before 5G).

6) Is labelled Figure 7 in the Figures. What is the light pink shape "attacking" the tethered vesicle?

We corrected this. The Figure number was changed to Figure 8 due to the addition of the electron microscopy (Figure 3) and GAP mutant (Figure 7) figures. The light pink shape was initially thought to display the possibility of Age2 to be recruited from the cytosol upon tethering. We removed this from the model, as we think it is much more likely that Age2 is already present on the vesicle, as can be judged by the strong enrichment in our TurboID assays.

Referee #2:

This study from Schoppe et al. addresses a few outstanding issues in the realm of AP-3 biology. The precise role of Vps41 in AP-3 trafficking has remained unclear. There is good evidence in the literature from the Ungermann group and others for Vps41 functioning as part of the HOPS complex to tether AP-3 vesicles to the lysosome/vacuole. However, an additional role of Vps41 in biogenesis of AP-3 vesicles has been proposed. This study addresses this potential role primarily through the use of an elegant anchor-away approach in which they ectopically localize Vps41 to mitochondria. Through these experiments, the authors provide convincing evidence that Vps41 is not required for AP-3 vesicle biogenesis but is sufficient for AP-3 vesicle tethering. They also show that the kinase Yck3 is important for vesicle tethering but not biogenesis. There are nice control experiments throughout the work, such as the requirement of the Apl5-ear domain for the anchor-away interaction between Vps41 and the AP-3 complex. In addition, they used a biotinylation (BioID) approach to identify proteins interacting with (or nearby) Apl5 in both w.t. and anchor-away cells. Using this approach they identified Age2 as an unexpected interactor. Age2 is an ArfGAP and a homolog of AGAP1, which has been implicated in

regulation of AP-3 in mammalian cells. The authors interpret the fact that Age2 is found to interact with Apl5 in both wt and anchor-away cells to mean that Age2 may be involved in the uncoating process. Importantly, they demonstrate that Age2 is not required for AP-3 vesicle biogenesis (in fact there are more vesicles formed in age2 mutants) and therefore Age2 is very likely to function to uncoat the AP-3 vesicle.

Overall I found this study to be very convincing and interesting. It also opens the field up for further studies investigating the precise mechanisms of vesicle uncoating.

We thank the reviewer for her/his positive feedback.

I have only minor suggestions for improvement.

1. On page 6 the authors cite a few previous studies using mitochondrial anchor-away to test protein-protein interactions. I think it would be good to also cite the work from Wei Guo's group where they anchored the exocyst subunit Sec3 to mitochondria and observed secretory vesicle tethering (Luo et al., MBoC 2014).

As suggested, we added the reference to the manuscript.

2. Figure panel 3A appears to not be specifically described in the results section?

Thank you for pointing this out, we added the required description in the text.

3. Circumstances permitting, it might be interesting to test whether the catalytic Arg residue of Age2 is required for Age2 function in their various AP-3 functional tests. This would provide more evidence for GAP activity as the key function of Age2 in AP-3 regulation. However, this is a relatively minor point and with research facilities shut down there is no reason to prevent publication without these experiments.

This is a very good suggestion. We generated strains carrying a plasmid with R52K (AGE2) or R54K (GCS1) mutations in the respective deletion backgrounds. We then screened for an AP-3 defect as judged by the GNS localization and in addition analyzed the numbers of Apl5 positive dots as done before for the deletion strains. Indeed, the GAP activity of Age2 appears to be critical for its function in AP-3 trafficking, as the mutant copies the phenotype of the deletion, which can be rescued by addition of WT AGE2. For Gcs1, we did not observe any changes in AP-3 trafficking in the GAP-dead mutant (see new Figure 7).

Referee #3:

This manuscript reports that the previously demonstrated interaction of the Vps41 subunit of HOPS with the Apl5 subunit of AP-3 is responsible for tethering AP-3 vesicles to the vacuole. This conclusion is in line with the function of the HOPS complex in vesicle tethering but contrasts with previous work suggesting that Vps41 functions as a clathrin-like coat for the formation of AP-3 vesicles. In addition, the authors report that the Age2 Arf1-GAP protein mediates AP-3-dependent sorting and probably uncoating of AP-3 vesicles. These conclusions are interesting, but, in my opinion, they don't rise to the level of novelty expected for a journal like EMBO J. Some of the findings presented in this paper have already been shown before, though in somewhat different ways. In addition, some of the results are difficult to appreciate, probably because of the challenge to visualize intracellular structures in yeast. Finally, there are some questions that are relevant but the manuscript leaves unanswered.

We thank the reviewer for her/his assessment. Naturally, we disagree with the final conclusion on previously shown aspects and relevance for the community. There has been an ongoing controversy on the role of Vps41 as a putative factor involved in vesicle formation that this deserved close inspection, and we show here that such a role of Vps41 is unlikely. We also uncovered that Age2 is an AP-3 specific GAP, which was also not known before. To further support our arguments, we now analyzed

the Age2 catalytic dead mutant, which inhibits AP-3 transport (new Figure 7) and show by immuno-EM that Vps41 can tether vesicles to mitochondria (Figure 3).

Specific points:

Are other subunits of the HOPS complex involved in tethering AP-3 vesicles to the vacuole or does Vps41 function independently of the other HOPS subunits? Vps39 KO also interferes with the AP-3 pathway. Is it also involved in tethering? Why isn't it detected in the BioID experiments? The role of HOPS and its individual subunits in tethering AP-3 vesicles is a fundamental issue that the study doesn't address.

Thank you for your questions. The involvement of HOPS is indeed a fundamental issue and has already been shown in a very nice study by the group of Alexey Merz (Angers and Merz, 2009). The HOPS complex requires all of its six subunits to function, and only then the AP-3 pathway is functional. One aim of this study was to answer the controversial question whether Vps41 is required for vesicle generation, and we here show that it is not needed in yeast.

Regarding the detection of HOPS subunits in the BioID experiments, we made a statement in the text that might not have been clear enough. The radius of biotinylation is approximately 10 nm. Apl5 interacts with the HOPS subunit Vps41, which is on one end of the elongated shape of HOPS, and close to this site is also Vps33 and Vps16. As we were able to detect Vps41 (Figure 5) and also Vps33 (though not with a significant score) we think, that this restriction in labeling area is the reason why we only find these two subunits. We rephrased and extended the sentence on page 10 to point this out in a better way.

Although KO of Age2 increases the number of AP-3 vesicles, this observation doesn't necessarily mean that it plays a role in uncoating; other scenarios are possible. More direct evidence for a role in uncoating is needed.

We agree that our data does not necessarily show that Age2 functions in uncoating, although judged by data for AP-1 and COPI it seems to be the most plausible explanation. We extended the discussion in this direction, taking also other roles of ArfGAPs (cargo enrichment) into account. Furthermore, we overexpress Age2 and Gcs1 under control of a TEF promoter, asking whether increased expression of the ArfGAP would cause premature AP-3 coat release and thus a block in transport, though we did not observe any defect (Figure 7A-D). However, we now show that a GAP-dead mutant of Age2, but not Gcs1, causes an AP-3 defect, providing further support for its direct role on AP-3 vesicles (Figure 7).

Fig. 1A,B. The difference in co-localization of Apl5 with Vps41 at the permissive and non-permissive temperature is statistically significant but very small (5%?). If there is a block in fusion, why isn't there a bigger shift toward the vacuole? And why doesn't the amount of Apl5 in the Sec7 compartment decrease?

We are very careful in our interpretation here. At present we see some difference, but if we consider the very small change, we do not expect a massive enrichment. Another problem here is that we do not know which fraction of Apl5 is present at the Golgi or vacuole at any time point, and as the reviewer rightly pointed out before, we look here at structures that are hard to track. We took this data as an entry into the study.

Fig. 1D,E. The independence of AP-3-mediated sorting from clathrin was already shown in previous studies, so these experiments add little to what is already known. With regards to the results, why does GNS accumulate at the plasma membrane in AP-3 mutants? Why is it not endocytosed (as mentioned in the text)? Is AP-3 involved in endocytosis?

It is correct, that the independence of clathrin has been suggested before. However, not in the way we do it in context of this manuscript, utilizing the GNS construct. Thus, we think this experiment provides

additional prove in this direction. As explained in the text, GFP-Nyv1-Snc1 is a fusion construct containing the cytoplasmic tail of Nyv1 carrying the AP-3 sorting signal and the TMD from Snc1 that cycles to the PM. This TMD is the reason why the cargo gets sorted to the plasma membrane when there is a defect in the AP-3 pathway. This construct does not contain an endocytic sorting motif and consequently endocytosis is rather slow as reported before, yet will reach the vacuole via the endocytic pathway. Since this is a well-established reporter for AP-3 trafficking that has been used in several studies, including ours (Cabrera et al., 2009, 2010, 2013), we use it here only to report on the AP-3 defect and its extent without addressing its precise sorting.

Fig. 2C. The results are very difficult to appreciate. I suggest that they show bigger pictures of the yeast cells.

Thank you for pointing this out. As suggested, we increased the size of the pictures.

Fig. 2F. What band is Nyv1? Differences in crude and pure mitochondrial fractions of both constructs are difficult to see. The text mentions a 3-fold difference that is not clear from these images.

The non-specific band is indicated by a white asterisk as pointed out in the figure legend. While the difference is not black and white, the difference in the pure mitochondrial fractions between the two different strains is clearly visible. Since there are still some vacuolar contaminations present in the crude fraction as judged by the Vma1 signal, we decided to add a second purification step to make sure that the Nyv1 signal corresponds to a vesicular signal in context of AP-3 (Figure 2F).

While there is a decrease in the Nyv1 signal for the Δ PEST mutant, we do not observe this for the S-D mutant, while the Vma1 signal decreases in both cases. This indicates that we co-purify a source of Nyv1 that is not vacuoles and consequently has to be AP-3 vesicles. We added a quantification of the Data to the Figure (now Figure 2G).

Figs. 3A and 3B are not described in the text. If they are not necessary, move them to a supplementary figure or delete.

Thank you for pointing this out, we added the required description to the text.

Fig. 3C. Make it larger so that differences can be better appreciated.

As suggested, we increased the size of the pictures.

Fig. 5F. The additional number of Apl5 dots in Age2 mutants is not clear in this panel.

Agreed. While in the whole picture one can see a difference, it is not so easy to appreciate this in the inset. We chose a different image that is more representative of the data.

Use "Inset" in place of "inlet" in the figures.

As suggested, we changed the description in the figures.

1st Revision - Editorial Decision

9th Jul 2020

Thank you for submitting a revised version of your manuscript. It has now been seen by two of the original referees, whose comments are shown below.

As you will see, they find that criticisms have been sufficiently addressed and recommend the study for publication. However, there are a few editorial issues concerning text and figures that I need you to address, before we can officially accept your manuscript.

Referee #2:

The authors have further strengthened their manuscript with new experimental data and textual revisions. In particular the new GAP dead mutant experiments comparing Age2 to Gcs1 turned out just as would be predicted. I think this work is very elegant and convincing and I strongly recommend publication.

Referee #3:

No further comments

2nd Authors' Response to Reviewers

16th Jul 2020

The authors performed the requested editorial changes.

2nd Revision - Editorial Decision

22nd Jul 2020

I am pleased to inform you that your manuscript has been accepted for publication in The EMBO Journal.

Corresponding Author Name: Christian Ungermann

Manuscript Number: EMBOJ-2020-105117RR